# Discrete Information Dynamics with Confidence via the Computational Mechanics Bootstrap: Confidence Sets and Significance Tests for Information-Dynamic Measures

**DOI:** 10.3390/e22070782

**Published:** 2020-07-17

**Authors:** David Darmon

**Affiliations:** Department of Mathematics, Monmouth University, West Long Branch, NJ 07764, USA; ddarmon@monmouth.edu

**Keywords:** information theory, information dynamics, computational mechanics, bootstrap, confidence distributions, confidence sets, significance tests

## Abstract

Information dynamics and computational mechanics provide a suite of measures for assessing the information- and computation-theoretic properties of complex systems in the absence of mechanistic models. However, both approaches lack a core set of inferential tools needed to make them more broadly useful for analyzing real-world systems, namely reliable methods for constructing confidence sets and hypothesis tests for their underlying measures. We develop the computational mechanics bootstrap, a bootstrap method for constructing confidence sets and significance tests for information-dynamic measures via confidence distributions using estimates of ϵ-machines inferred via the Causal State Splitting Reconstruction (CSSR) algorithm. Via Monte Carlo simulation, we compare the inferential properties of the computational mechanics bootstrap to a Markov model bootstrap. The computational mechanics bootstrap is shown to have desirable inferential properties for a collection of model systems and generally outperforms the Markov model bootstrap. Finally, we perform an in silico experiment to assess the computational mechanics bootstrap’s performance on a corpus of ϵ-machines derived from the activity patterns of fifteen-thousand Twitter users.

## 1. Introduction

Outside of the physical sciences, much of the scientific process involves model building from empirical observations. For systems evolving in time, this model building typically involves identifying a proxy stochastic process that, at least approximately, results in realizations that closely match the data at hand. Information dynamics provide a set of measures for analyzing such a stochastic process, by quantifying how the process stores, processes, and transmits information when viewed as a communication channel from its past through its present to its future. Computational mechanics subsumes information dynamics and reveals the complete computational structure of the system via its ϵ-machine representation. Together, these two approaches comprise a toolbox for analyzing time series viewed as realizations of stochastic processes, and they have been applied to physical [1,2,3], biological [4,5,6], social [7,8,9], and engineered/artificial systems [10,11,12,13].

Information dynamics and computational mechanics are well-developed in theory. However, more work must be done to move from summarizing the properties of an available time series to making inferences about the underlying process that generated the time series. This is the move from descriptive statistics, which are fairly well-developed for information dynamics, to statistical inference. From a simplified view, the three main tasks of statistical inference are point estimation, interval estimation, and hypothesis testing. A point estimate provides a single numerical value (or more generally, a single element from a set) that best approximates some property of a stochastic process. An interval estimate provides an interval of values (or more generally, a set of values), such that the property of the stochastic process falls in that set with some prespecified probability. A hypothesis test provides a tentative answer to whether a property of the stochastic process equals some value (or more generally, falls into some set of values). Of these three tasks, point estimation has been the main focus in the information dynamics community, which has developed various estimators, often plug-in, for information-dynamic quantities. The other two tasks, interval estimation and hypothesis testing, have far fewer tools available, with some notable exceptions. In [14], the authors develop a Markov Chain Monte Carlo method to construct Bayesian confidence intervals (really, credible intervals) for the entropy rate of a process using a context tree weighting procedure, and later generalize this approach to construct credible intervals for average and specific mutual information rate [15]. The authors of [16] present a Bayesian procedure for determining a posterior distribution over a certain class of ϵ-machines that could, in principle, be used to determine posterior distributions over information-dynamic measures. In [17], the authors developed a bootstrapping procedure to construct confidence intervals for information-dynamic properties of continuous-state systems using echo state network simulators.

The three tasks of inferential statistics can all be easily accomplished using the machinery of confidence distributions. A confidence distribution provides a frequentist summary of all the inferences that can be made from the available data and can be thought of the frequentist answer to the Bayesian posterior distribution [18,19,20]. With a confidence distribution in hand, one may construct point and interval estimates and perform hypothesis tests by simple inspection of the confidence distribution.

In this paper, we adapt the machinery of confidence distributions to make statistical inferences about the information- and computation-theoretic properties of discrete-state stochastic processes evolving in discrete-time. We develop the computational mechanics bootstrap, a method for constructing bootstrap confidence distributions for information-dynamic measures from a realized time series. The computational mechanics bootstrap uses an ϵ-machine simulator to generate bootstrap time series from a stochastic process and uses those bootstrap time series to create a bootstrap distribution of estimates that may be used directly or further refined to use as a confidence distribution. These bootstrap confidence distributions can then be used to make inferences regarding a given information-dynamic quantity.

The rest of the paper is organized, as follows. In Section 2, we review information dynamics, computational mechanics, and their associated measures. We then review methods for model inference and model selection in the context of computational mechanics, as well as various plug-in estimators for information- and computation-theoretic quantities. Next, we describe the machinery of confidence distributions and their use for statistical inference. We then develop the computational mechanics bootstrap, which constructs confidence distributions from time series bootstrapped from an inferred ϵ-machine. In Section 3.1 we conduct a simulation study in order to validate the inferential properties of confidence distributions derived from the computational mechanics bootstrap and compare the computational mechanics bootstrap to a bootstrap based on a Markov model simulator. In Section 3.2, we then apply the computational mechanics bootstrap to a collection of 14,427 stochastic processes derived from the user activity of 15,000 Twitter accounts to further investigate the inferential properties of the computational mechanics bootstrap with a diverse, real-world data set. In Section 4, we consider the limitations of the proposed version of the computational mechanics bootstrap and recommend further refinements to the method. We conclude in Section 5 by reviewing the main findings and our general recommendations for using the computational mechanics bootstrap in applied work.

## 2. Methods

### 2.1. Notation and Conventions

In this paper, we consider discrete-state stochastic processes evolving in discrete time. We will denote the output of the process at time *t* by $X_{t}$, where a realization $x_{t}$ of $X_{t}$ is an element of the finite alphabet X. We will denote a consecutive sequence of outputs from the process by $X_{a}^{b}=\left(X_{a},X_{a+1},\ldots ,X_{b-1},X_{b}\right)$, and the semi-infinite past and future relative to time *t* by $X_{-\infty }^{t-1}=\left(\ldots ,X_{t-2},X_{t-1}\right)$ and $X_{t}^{\infty }=\left(X_{t},X_{t+1},\ldots \right)$, respectively. Moreover, we will assume that the process is conditionally stationary [21], such that $P\left(X_{t}^{\infty }|X_{-\infty }^{t-1}=x\right)=P\left(X_{0}^{\infty }|X_{-\infty }^{1}=x\right)$ for all possible semi-infinite pasts x and times *t*, and will therefore drop the dependence on *t*. In the following, we assume familiarity with information theory at the level of [22].

### 2.2. Information Dynamics and Computational Mechanics

There is a rich repertoire of measures from the field of information dynamics [23,24]. We focus in this paper on inference for three of the most commonly used: the entropy rate, excess entropy, and statistical complexity of a process. The entropy rate $h_{\mu }$ of a process quantifies the uncertainty about its next-step future $X_{0}$ conditional on its past $X_{-L}^{-1}$ in the limit of an infinitely long past,
(1)$$\begin{array}{cc} h_{\mu } & =H[X_{0}|X_{-\infty }^{-1}]\\ & ≜\lim_{L\to \infty }H\left[X_{0}|X_{-L}^{-1}\right] \end{array}$$
where H[·∣·] is the usual conditional Shannon entropy of the future $X_{0}$ given the past $X_{-L}^{-1}$. The entropy rate captures the residual uncertainty remaining about a process’s future after accounting for its entire past and, thus, measures the intrinsic randomness in a process. For example, a deterministic process has an entropy rate of 0 bits, while a Bernoulli process has an entropy rate of 1 bit.

The excess entropy E of a process is the mutual information between a process’s past and its future,
(2)$$\begin{array}{c} E=I[X_{-\infty }^{-1};X_{0}^{\infty }], \end{array}$$
and, thus, quantifies the reduction in the uncertainty about the entire future upon knowing the entire past and vice versa [23,25,26]. Excess entropy is equivalently defined as the cumulative reduction in uncertainty from considering pasts of length *L* relative to considering a semi-infinite past,
(3)$$\begin{array}{cc} E & =\sum_{L=1}^{\infty }\left(H\left[X_{0}|X_{-L}^{-1}\right]-H\left[X_{0}|X_{-\infty }^{-1}\right]\right)\\ & =\sum_{L=1}^{\infty }\left(h_{\mu }\left(L\right)-h_{\mu }\right), \end{array}$$
which motivates the name “excess” entropy. From this perspective, excess entropy quantifies the cumulative cost of forgetting longer pasts. For finitary processes, i.e., those with a finite number of causal states, the entropy rate and excess entropy are related, asymptotically, by
(4)$$\begin{array}{c} H\left[X_{1}^{L}\right]\propto E+h_{\mu }L, \end{array}$$
so that the entropy rate gives the rate of growth of the *L*-block entropies and the excess entropy specifies the overall offset.

While information dynamics provides model-free measures of the information-theoretic properties of a stochastic process, computational mechanics provides a constructive representation of any stationary stochastic process in terms of a Hidden Markov Model intrinsic to the process itself [27,28]. This model is known as the ϵ-machine for the process, and its hidden states are known as the causal states of the process. Unlike a standard Hidden Markov Model, computational mechanics provides a constructive definition of the ϵ-machine for a given process. The hidden states are defined via an equivalence relation over semi-infinite pasts such that the predictive distributions over semi-infinite futures are identical. That is, two pasts $u_{-\infty }^{-1}$ and $v_{-\infty }^{-1}$ are equivalent if and only if
(5)$$\begin{array}{c} P\left(X_{0}^{\infty }|X_{-\infty }^{-1}=u_{-\infty }^{-1}\right)=P\left(X_{0}^{\infty }|X_{-\infty }^{-1}=v_{-\infty }^{-1}\right). \end{array}$$

The mapping from a past $u_{-\infty }^{-1}$ to its corresponding equivalence class is typically denoted by ϵ, so the equivalence relation is given by $u_{-\infty }^{-1}{∼}_{\epsilon }v_{-\infty }^{-1}$, and the set of causal states (equivalently, the set of equivalence classes) is denoted by S.

The mapping ϵ induces the hidden state process $\left(\ldots ,S_{-1},S_{0},S_{1},\ldots \right)$ via $S_{t}=\epsilon \left(X_{-\infty }^{t}\right)$, which is called the causal state process. The causal state process has several desirable properties, including causally shielding the past of the observable process from its future and being Markov, even when the original process is not. For our purposes, the ϵ-machine and causal state process have the useful property that many information-dynamic measures can be computed from the ϵ-machine in closed-form, as we will review in Section 2.4.

The ϵ-machine also introduces a new information-dynamic measure intrinsic to a stochastic process, its statistical complexity. The statistical complexity of a stochastic process is defined as the mutual information between the semi-infinite past $X_{-\infty }^{-1}$ and the associated causal state $S_{-1}=\epsilon \left(X_{-\infty }^{-1}\right)$,
(6)$$\begin{array}{c} C_{\mu }=I\left[S_{-1};X_{-\infty }^{-1}\right]. \end{array}$$

Because the causal state $S_{-1}$ is predictively sufficient for the future $X_{0}^{\infty }$, the statistical complexity also corresponds to the average amount of information regarding the process’s past necessary to optimally predict its future and, in this sense, quantifies the complexity of the process. The excess entropy of a process places a lower bound on the statistical complexity of a process, $C_{\mu }\geq E$.

### 2.3. Model Inference and Model Selection

In what follows, we develop a method for bootstrapping confidence distributions for information- dynamic measures from an inferred ϵ-machine. In this paper, we estimate the ϵ-machine using the Causal State Splitting Reconstruction (CSSR) algorithm [29]. However, other methods of inferring ϵ-machine could be used, including those that are based on topological methods [30], spectral methods [31], robust causal states [32], and integer programming [33]. Similar investigations of these methods as done in this paper would help to elucidate the strengths and weaknesses of each method for bootstrapping confidence distributions, which we defer to future work.

The CSSR algorithm estimates an ϵ-machine by starting with a Bernoulli process, the simplest possible model for a stochastic process, and then adding structure via additional causal states, as warranted by hypothesis tests for differences in the past-conditional predictive distributions, and finally determinizing the expanded states to produce an ϵ-machine. The CSSR algorithm yields an estimator for the underlying ϵ-machine, which is known to converge in probability as long as the observable process (a) is conditionally stationary, (b) is finitary, and (c) there exists a length Λ<∞, such that every causal state can be synchronized to by at least one past $x_{-L}^{-1}$ with L≤Λ [34].

The CSSR algorithm has two tuning parameters that must be set prior to its use: the significance level α used for all of the hypothesis tests related to state-splitting and the maximum history length $L_{\max}$, which determines the longest pasts to consider during the splitting process. For all uses of CSSR in this paper, we fix the significance level at α=0.001. We treat the selection of $L_{\max}$ as a model order selection problem, and choose the value of $L_{\max}$ that minimizes Schwarz’s Bayesian Information Criterion (BIC) [35]. BIC is consistent for selecting the model order of a Markov process [36,37], and the causal state process underlying any conditionally stationary process is itself a Markov process.

The BIC for the estimated ϵ-machine ${\hat{\epsilon }}_{L}$ using a maximum lookback of *L* is given by
(7)$$\begin{array}{c} \mathrm{BIC}\left(L\right)=-2\log P_{{\hat{\epsilon }}_{L}}\left(X_{1}^{T}=x_{1}^{T}\right)+\dim\left({\hat{\epsilon }}_{L}\right)\cdot \log T \end{array}$$
where the dimension of the estimated ϵ-machine $\dim\left({\hat{\epsilon }}_{L}\right)=\left(|X|-1)\cdot \right|\hat{S}|$ is the number of transitions that must be estimated for the ϵ-machine [4]. The likelihood $P_{{\hat{\epsilon }}_{L}}\left(X_{0}^{T}=x_{1}^{T}\right)$ can be computed using the causal shielding property of the causal states via
(8)$$\begin{array}{cc} P_{{\hat{\epsilon }}_{L}}\left(X_{1}^{T}=x_{1}^{T}\right) & =\sum_{s_{0}\in S}P_{{\hat{\epsilon }}_{L}}\left(S_{0}=s_{0}\right)P_{{\hat{\epsilon }}_{L}}\left(X_{1}^{T}=x_{1}^{T}|S_{0}=s_{0}\right)\\ & =\sum_{s_{0}\in S}P_{{\hat{\epsilon }}_{L}}\left(S_{0}=s_{0}\right)\prod_{t=1}^{T}P_{{\hat{\epsilon }}_{L}}\left(X_{t}=x_{t}|S_{t-1}=s_{t-1}\right). \end{array}$$

For long time series, this form of the likelihood can become numerically unstable, since the product term may result in probabilities that underflow with floating point arithmetic. In the case that the causal state sequence can be determined after $L_{\mathrm{synch}}$ time steps, the likelihood can equivalently be factored as
(9)$$\begin{array}{cc} P_{{\hat{\epsilon }}_{L}}\left(X_{1}^{T}=x_{1}^{T}\right) & =P_{{\hat{\epsilon }}_{L}}\left(X_{1}^{L_{\mathrm{synch}}}=x_{1}^{L_{\mathrm{synch}}}\right)P_{{\hat{\epsilon }}_{L}}\left(X_{L_{\mathrm{synch}}+1}^{T}=x_{L_{\mathrm{synch}}+1}^{T}|X_{1}^{L_{\mathrm{synch}}}=x_{1}^{L_{\mathrm{synch}}}\right)\\ & =P_{{\hat{\epsilon }}_{L}}\left(X_{1}^{L_{\mathrm{synch}}}=x_{1}^{L_{\mathrm{synch}}}\right)\prod_{t=L_{\mathrm{synch}}+1}^{T}P_{{\hat{\epsilon }}_{L}}\left(X_{t}=x_{t}|S_{t-1}=s_{t-1}\right) \end{array}$$
where the first term in the factorization is computed via (Equation 8).

A common method for estimating the entropy rate of a process uses differences of entropies of blocks of symbols, which implicitly assumes a Markov model for the process. Therefore, we will compare the computational mechanics bootstrap to a bootstrap based on a Markov model simulator. For a Markov model, the only tuning parameter is the model order of the Markov chain, which we again select using the BIC based on the conditional likelihood of the Markov model,
(10)$$\begin{array}{c} \mathrm{BIC}\left(L\right)=-2\log P_{L}\left(X_{L+1}^{T}=x_{L+1}^{T}|X_{1}^{L}=x_{1}^{L}\right)+\left(|X\right|-{1)|X|}^{L}\cdot \log T \end{array}$$
where $P_{L}\left(X_{L+1}^{T}=x_{L+1}^{T}|X_{1}^{L}=x_{1}^{L}\right)$ factors according to the Markov model as
(11)$$\begin{array}{c} P_{L}\left(X_{L+1}^{T}=x_{L+1}^{T}|X_{1}^{L}=x_{1}^{L}\right)=\prod_{t=L+1}^{T}P_{L}\left(X_{t}=x_{t}|X_{t-L}^{t-1}=x_{t-L}^{t-1}\right). \end{array}$$

While we select the model order of the Markov model using BIC for direct comparison with the computational mechanics bootstrap, more sophisticated methods of Markov model order selection are available [38].

For both the ϵ-machine and the Markov models, we consider all the values of *L* from 1 to
(12)$$\begin{array}{c} L_{\max}=\frac{\log_{2}T}{\log_{2}\left|X\right|}-1 \end{array}$$
using the result of [39] that the distributions $P(X_{1}^{L})$ can only be consistently estimated for functions of a Markov chain when *L* scales like $\log_{2}T/h_{\mu }$ and we take $\log_{2}\left|X\right|$ as a crude upper bound for $h_{\mu }$.

### 2.4. Information- and Computation-Theoretic Estimators from the Inferred ϵ-Machine

The entropy rate, excess entropy, and statistical complexity of a stochastic process can all be derived from the ϵ-machine representation of the process. There are typically many routes from the ϵ-machine to the information- and computation-theoretic measures. The entropy rate and statistical complexity can be directly computed from the ϵ-machine and its stationary distribution P(S) over the causal states. Because the causal state at the most recent past is predictively sufficient for the future, it follows that
(13)$$\begin{array}{c} \begin{array}{cc} h_{\mu } & =H[X_{0}|X_{-\infty }^{-1}]\\ \; & =H[X_{0}|S_{-1}]\\ \; & =-\sum_{s_{-1}\in S}P\left(S_{-1}=s_{-1}\right)\sum_{x_{0}\in X}P\left(X_{0}=x_{0}|S_{-1}=s_{-1}\right)\log_{2}P\left(X_{0}=x_{0}|S_{-1}=s_{-1}\right). \end{array} \end{array}$$

Similarly, the statistical complexity of the ϵ-machine is given by the Shannon entropy of the stationary distribution of the causal states,
(14)$$\begin{array}{c} \begin{array}{cc} C_{\mu } & =I[S_{-1}\wedge X_{-\infty }^{-1}]\\ \; & =H\left[S_{-1}\right]=-\sum_{s_{-1}\in S}P\left(S_{-1}=s_{-1}\right)\log_{2}P\left(S_{-1}=s_{-1}\right). \end{array} \end{array}$$

The excess entropy can be directly calculated from either a bidirectional ϵ-machine for the process [40,41] or a spectral representation of the ϵ-machine [42]. We approximate the excess entropy via a truncation of the cumulative deviation of the *L*-back predictive entropies from the entropy rate,
(15)$$\begin{array}{cc} E & =I[X_{-\infty }^{-1}\wedge X_{0}^{\infty }]\\ & =\sum_{L=1}^{\infty }\left(H\left[X_{0}|X_{-L}^{-1}\right]-H\left[X_{0}|X_{-\infty }^{-1}\right]\right)\\ & =\sum_{L=1}^{\infty }\left(h_{\mu }\left(L\right)-h_{\mu }\right)\\ & \approx \sum_{L=1}^{L_{\mathrm{tr}}}\left(h_{\mu }\left(L\right)-h_{\mu }\right), \end{array}$$
where the *L*-block entropies can be computed from the mixed state presentation of the ϵ-machine [43]. We truncate the sum at $L_{\mathrm{tr}}=200$, which, for processes with relatively short memory, is sufficient for approximating the full excess entropy.

For the entropy rate, statistical complexity, and excess entropy, we use plug-in estimates where we compute these measures from the estimated ϵ-machine inferred by applying CSSR to the time series. For a given time series $X_{1}^{T}$ with ϵ-machine estimate $\hat{\epsilon }$, this results in the point estimates ${\hat{h}}_{\mu }$, ${\hat{C}}_{\mu }$, and $\hat{E}$. Because an order *L* Markov model is just a special case of an ϵ-machine where each *L*-length past corresponds to its own causal state, the plug-in estimates from the estimated Markov model are also computed via plug-in estimates.

### 2.5. Confidence Distributions and Their Use for Inference

Consider a generic property θ of the stochastic process, where, in our case, θ will be one of the entropy rate, statistical complexity, or excess entropy. To construct confidence intervals and hypothesis tests for θ, we use the theory of confidence distributions [18,19,20]. A confidence distribution $C\left(\theta \right)\equiv C\left(\theta ;X_{1}^{T}\right)$ is a function of both the parameter and the data, such that (a) for any given $X_{1}^{T}$, $C(\cdot ;X_{1}^{T})$ is a cumulative distribution function with respect to θ and (b) at the true parameter value $\theta_{0}$, $C(\theta_{0};\cdot )$ follows a continuous uniform distribution on the interval [0,1]. Condition (a) guarantees the confidence distribution is a distribution over θ, and condition (b) guarantees that the confidence distribution results in valid *p*-values and, given the duality between hypothesis testing and interval estimation, confidence intervals for θ. Confidence distributions can be constructed via pivots. A pivot is a function of both the sample and the parameter of interest whose sampling distribution is known exactly and does not depend on the parameter. The classic example is the *t*-statistic $t=\sqrt{n}\left(\bar{X}-\mu \right)/S$ for the mean μ of a Gaussian population, where $\bar{X}$ and *S* are the mean and standard deviation of a random sample of size *n* from the population, in which case *t* follows the *t*-distribution with n−1 degrees of freedom. However, a pivot for a generic property cannot always be determined. In the absence of a pivot, condition (b) is weakened to (b*_a_*) at the true parameter value $\theta_{0}$, $C(\theta_{0};\cdot )$ converges in distribution to a uniform distribution on the interval [0,1] as *T* goes to infinity. This results in an asymptotic confidence distribution that produces asymptotically valid *p*-values and confidence intervals. We will only consider asymptotic confidence distributions in this paper, and, thus, drop the modifier asymptotic in our presentation.

A confidence distribution can be used to perform all of the standard inferential tasks regarding a parameter and, in this sense, is similar to a Bayesian posterior distribution for the parameter, except without specification of a prior distribution for the parameter. Viewed as a cumulative distribution function for θ, C(θ) assigns an epistemic probability to the interval (−∞,θ]. For this reason, confidence distributions can be directly used to determine the *p*-value for a particular hypothesis test. In fact, the confidence distribution is equivalent to the curve traced out by the *p*-value for a right-sided test as the null value $\theta_{0}$ varies. Thus, the *p*-value for a right-sided test
(16)$$\begin{array}{c} \begin{array}{cc} H_{0} & :\theta \leq \theta_{0}\\ H_{1} & :\theta >\theta_{0} \end{array} \end{array}$$
is given by $C(\theta_{0})$, and the *p*-value for the corresponding left-sided test is given by $1-C(\theta_{0})$. The *p*-value for the two-sided test
(17)$$\begin{array}{c} \begin{array}{cc} H_{0} & :\theta =\theta_{0}\\ H_{1} & :\theta \neq \theta_{0} \end{array} \end{array}$$
is given by $2\min\{C\left(\theta_{0}\right),1-C\left(\theta_{0}\right)\}$. Figure 1a shows a schematic of a realization of a generic confidence distribution for $h_{\mu }$, as well as the one-sided and two-sided *p*-values for a particular null value $h_{\mu ,0}$ for $h_{\mu }$.

Confidence distributions can also be used to construct confidence sets for a parameter. For example, a two-sided, equi-tailed confidence interval with coverage (“confidence level”) 1−α is defined by the interval $\left(C^{-1}\left(\alpha /2\right),C^{-1}\left(1-\alpha /2\right)\right)$, where $C^{-1}$ is the quantile function for the confidence distribution *C*. A more direct route to constructing confidence intervals is via the confidence curve cc(θ) for the parameter,
(18)$$\begin{array}{c} \mathrm{cc}(\theta )=|2C(\theta )-1|. \end{array}$$

A confidence curve is equivalent to the curve that is traced out by the left- and right-endpoints of a two-sided confidence interval as the coverage probability 1−α varies from 0 to 1. Figure 1b shows a schematic of a realization of the confidence curve derived from the confidence distribution in Figure 1a. The 95% confidence interval is highlighted in green, corresponding to the sublevel set $\left\{h_{\mu }:\mathrm{cc}\left(h_{\mu }\right)\leq 0.95\right\}$.

### 2.6. Bootstrapping Confidence Distributions from ϵ-Machines

We bootstrap from the estimated ϵ-machine in order to construct a confidence distribution for a given information-dynamic measure θ. To bootstrap from the ϵ-machine, we repeatedly sample time series $x_{1}^{T*}$ according to the estimated ϵ-machine. Let ${\hat{\epsilon }}_{L_{0}}$ be the ϵ-machine estimated from the original time series $x_{1}^{T}$, with model order $L_{0}$ chosen to minimize BIC. To sample each $x_{1}^{T*}$, we begin by choosing an initial causal state $s_{0}^{*}$ by sampling according to the stationary distribution $P_{{\hat{\epsilon }}_{L_{0}}}\left(S_{0}\right)$ over the causal states. The realization of the time series as well as its causal state series are then generated by sampling $x_{t}^{*}$ from $P_{{\hat{\epsilon }}_{L_{0}}}\left(X_{t}|S_{t-1}=s_{t-1}^{*}\right)$ for t=1,…,T and updating the new causal state via $s_{t}^{*}={\hat{\epsilon }}_{L_{0}}\left(s_{t-1}^{*},x_{t}^{*}\right)$. We then repeat this process *B* times. We apply CSSR to each bootstrap time series $x_{1}^{T*}$, resulting in a new ϵ-machine estimate ${\hat{\epsilon }}_{L_{0}}^{*}$, for which the bootstrap plug-in estimate for the measure is then ${\hat{\theta }}^{*}$. Note that we use the same $L_{0}$ both to generate the bootstrap time series and to estimate the ϵ-machines from the bootstrap time series. Alternatively, we could select a new $L_{0}^{*}$ for each bootstrap time series $x_{1}^{T*}$, again using BIC. This, of course, would be more computationally expensive, requiring an additional round of model selection for each of the *B* bootstrap time series.

The bootstrap distribution is given by the empirical distribution $\hat{F}\left(\theta \right)$ of the bootstrap estimates
(19)$$\begin{array}{c} \hat{F}\left(\theta \right)=\frac{1}{B}\sum_{b=1}^{B}I\left[{\hat{\theta }}_{b}^{*}\leq \theta \right] \end{array}$$
which can be directly taken as the percentile bootstrap confidence distribution
(20)$$\begin{array}{c} C_{\mathrm{pb}}\left(\theta \right)=\hat{F}\left(\theta \right). \end{array}$$

The bias-corrected bootstrap confidence distribution is a simple modification of the percentile bootstrap confidence distribution given by
(21)$$\begin{array}{c} C_{\mathrm{bcb}}=\Phi \left(\Phi^{-1}\left(\hat{F}\left(\theta \right)\right)-2b\right) \end{array}$$
where $b=\hat{F}\left(\hat{\theta }\right)$ and Φ is the cumulative distribution function of a standard Gaussian (“Normal”) random variable. The bias-corrected bootstrap confidence distribution may adjust for the bias in the estimate of θ by accounting for the empirical bias between $\hat{\theta }$ and the bootstrap distribution of ${\hat{\theta }}^{*}$. Either bootstrap confidence distribution may then be used as described in the previous section to perform hypothesis tests or construct confidence intervals for θ.

The computational mechanics bootstrap using the percentile bootstrap confidence distribution is summarized in Box 1.

Box 1The Computational Mechanics Bootstrap.**Input:** A time series $x_{1}^{T}$ from a discrete-state, discrete-time stochastic process.**Output:** A confidence distribution C(θ) for a measure θ.
Construct the ϵ-machines ${\left\{{\hat{\epsilon }}_{L}\right\}}_{L=1}^{L_{\max}}$ from $x_{1}^{T}$ using CSSR, where $L_{\max}=\frac{\log_{2}T}{\log_{2}\left|X\right|}-1$.Select the ϵ-machine ${\hat{\epsilon }}_{L_{0}}$ that minimizes the BIC (Equation 7).Compute $\hat{\theta }$ from ${\hat{\epsilon }}_{L_{0}}$.For b=1,…,B:
(a)Generate the time series $x_{1}^{T*}$ from ${\hat{\epsilon }}_{L_{0}}$.(b)Construct the ϵ-machine ${\hat{\epsilon }}_{L_{0}}^{*}$ from $x_{1}^{T*}$ using CSSR.(c)Compute ${\hat{\theta }}_{b}^{*}$ from ${\hat{\epsilon }}_{L_{0}}^{*}$Construct the confidence distribution $C_{\mathrm{pb}}\left(\theta \right)=\frac{1}{B}\sum_{b=1}^{B}I\left[{\hat{\theta }}_{b}^{*}\leq \theta \right].$


## 3. Results

### 3.1. Simulation Study

We begin with a simulation study to explore the inferential properties of the bootstrap confidence distributions constructed using either Markov or ϵ-machine simulators. We proceed from simple to increasingly complex stochastic processes, beginning with a renewal process and an alternating renewal process, both of which are Markov, and then considering the even process, which is not Markov, and the simple nonunifilar source, which is not finitary. The ϵ-machines for the four systems that are considered in this section are shown in Figure 2.

For each process, we generate S= 2000 time series from the process, and for each time series, we generate B=2000 bootstrap time series via both the Markov and ϵ-machine simulator. We then construct the percentile bootstrap and bias-corrected bootstrap confidence distributions to compute *p*-values and confidence intervals. We consider time series of lengths T= 1000 and T= 10,000. The entropy rate, excess entropy, and statistical complexity for all of the processes considered in the simulation study are given in Table 1.

#### 3.1.1. Renewal Process

The first process that we consider is a renewal process. A discrete-time renewal process is a stochastic process that is completely determined by the distribution over the number of time steps between adjacent emissions of a 1. Renewal process models are especially popular for modeling social [44,45,46,47,48] and neuroscientific [49,50] point processes, amongst many others. Renewal processes have stereotyped ϵ-machine architectures, cataloged in [51]. The renewal process considered here has a unique start state, labeled A in the ϵ-machine in Figure 2, and it eventually transitions to a unique end state, labeled C, on emissions of 0, with all emissions of 1 leading back to the start state. The particular renewal process used here was taken from a collection of models of user behavior on Twitter, which we will discuss in Section 3.2. This renewal process has three states, and thus is also a Markov process of order 2. However, it is not the most general second-order Markov process, since $P\left(X_{0}|X_{-2}^{-1}=\left(0,1\right)\right)=P\left(X_{0}|X_{-2}^{-1}=\left(1,1\right)\right)$, so it has only three, rather than four, causal states.

We first consider the distribution of *p*-values for the two-sided test (Equation 17), which are calculated from the confidence distributions via $P=2\min\{C\left(\theta_{0}\right),1-C\left(\theta_{0}\right)\}$. We evaluate the *p*-values, where $\theta_{0}$ is the true value of the parameter, as given in Table 1. In this case, the *p*-value should be uniformly distributed on [0,1]. Figure 3 shows the empirical distribution of the *p*-values across 2000 simulations. We see that the *p*-values for $h_{\mu }$ are nearly uniformly distributed for both the Markov and computational mechanics bootstraps for both T= 1000 and T= 10,000 length time series, with the *p*-values from the bias-corrected bootstrap confidence distribution performing slightly better at small levels of significance. Similar results hold for the *p*-values for E. The *p*-values for $C_{\mu }$ are stochastically smaller than the uniform distribution when T= 1000 for both types of bootstraps. This results in an inflated Type I Error Rate at all significance levels, since the actual probability of rejecting the null hypothesis when it is true is greater than the nominal probability. The computational mechanics bootstrap performs slightly better, i.e., having a smaller deviation of the Type I Error The rate from the desired level α. When T= 10,000, the computational mechanics bootstrap results in *p*-values that are stochastically greater than the uniform distribution, while the Markov model bootstrap continues to have an inflated Type I Error Rate for all values of α. This is due to the fact that, while the 3-state renewal process is a Markov model of order 2, the estimator from the Markov model does not account for the fact that two of the four possible causal states are, in fact, equivalent, and this inflates the estimate of the statistical complexity from the Markov model relative to its true value.

We next consider the power of each of the bootstrap hypothesis tests to detect a discrepancy from the null hypothesis for the test (Equation 17). The power of a hypothesis test is the probability that the null hypothesis is rejected. Fixing the significance level at α=0.35, we consider the proportion of simulated *p*-values that are less than or equal to α as we vary the null values $\theta_{0}$ away from the true values given in Table 1. The power of the hypothesis tests for each of the three measures is shown in Figure 4 as the null value used in the hypothesis test deviates from the true value. In the ideal case, the power should be less than or equal to 0.05 at the true values of the measures, and approach 1 for either positive or negative deviations from the true value. As expected, at a given deviation from the true value, the tests become more powerful as a longer time series is used. The tests based on the computational mechanics bootstrap are generally as powerful as the tests that are based on the Markov model bootstrap, except for testing small values of $h_{\mu }$, E, and $C_{\mu }$. *p*-values from the percentile bootstrap and bias-corrected bootstrap confidence distributions result in similar power, with the bias-corrected bootstrap confidence distributions having slightly better power for testing small values of $h_{\mu }$, E, and $C_{\mu }$.

Finally, we consider the coverage properties of confidence intervals constructed while using the bootstrap distributions. At a given nominal coverage probability 1−α, we compute the proportion of confidence intervals across the S= 2000 simulations that capture the true value of the measures given in Table 1, i.e., the empirical coverage probability. Figure 5 shows the difference between the empirical coverage probabilities and the nominal coverage probabilities, as well as pointwise 95% confidence intervals for the differences in coverage. A positive deviation indicates that the confidence intervals overcover and are, therefore, conservative, while a negative deviation indicates that the confidence intervals undercover are therefore anticonservative. Typically, a conservative confidence interval is preferred to an anticonservative confidence interval, since then at least the desired coverage probability is attained.

We see that, for both $h_{\mu }$ and E, both the Markov model and computational mechanics bootstraps result in confidence intervals that either slightly undercover (when T= 1000) or slightly overcover (when T= 10,000). In contrast, for $C_{\mu }$, both bootstraps lead to undercoverage when T= 1000, while the undercoverage persists for the Markov model bootstrap, even when T= 10,000. Again, this is due to the fact that the Markov model treats all four candidate causal states as distinct, leading to systematic bias in the estimate of $C_{\mu }$ via the Markov model.

#### 3.1.2. Alternating Renewal Process

We next consider the second process from Figure 2, an alternating renewal process. Alternating renewal processes are a generalization of renewal processes, where both the distribution over run lengths of 0 s and 1 s can be specified. Like renewal processes, alternating renewal processes have stereotyped ϵ-machine architectures, with unique start states for both the first 0 in a sequence of 0 s and the first 1 in a sequence of 1 s [9]. The particular alternating renewal process used here was again taken from the Twitter accounts discussed in Section 3.2. This alternating renewal process is equivalent to a generic second-order Markov process, with a single causal state for each of the four possible histories of length 2. Thus, the computational mechanics bootstrap can, in principle, perform no better than the Markov bootstrap for this system, since the process is a Markov model, and the CSSR algorithm must perform more inferential work both by discovering the correct ϵ-machine architecture and inferring its transition probabilities. This contrasts with the previous example, where the computational mechanics bootstrap had a potential advantage due to the additional structure in the process not captured by a full Markov model.

For the sake of brevity, for this and the remaining processes, we only consider the coverage properties of the confidence intervals constructed while using the bootstrapped confidence distributions. Figure 6 shows the deviations of the empirical coverage probabilities from the nominal coverage probabilities. As with the renewal process, we see that, for $h_{\mu }$ and E, both the Markov model and computational mechanics bootstraps produce confidence intervals that attain the desired coverage. However, the confidence intervals from the Markov model bootstrap outperform the computational mechanics bootstrap in capturing $C_{\mu }$, with the Markov model bootstrap confidence intervals attaining the desired coverages while the computational mechanics bootstrap confidence intervals undercover at all of the nominal coverage probabilities considered. The undercoverage improves when the initial model is inferred from a longer time series, but is still appreciable. Again, this is a process where the computational mechanics bootstrap can perform no better than the Markov model bootstrap, and we see that it performs worse at capturing the statistical complexity due to the additional statistical effort needed to discover the correct ϵ-machine architecture.

#### 3.1.3. Even Process

As our next example, we consider the even process. The even process is a strictly sophic process [52,53]: while the even process has a finite number of causal states, it is not representable by a Markov model of any finite order. The even process corresponds to a stochastic process where runs of 1 s of odd length are forbidden, and runs of 11 s and 0 s follow geometric distributions. Because the even process is strictly sophic, we expect the confidence intervals constructed while using the Markov model bootstrap to break down. For example, it is known that Markov models of order 10 or higher are necessary to approximate the the entropy rate of the even process [38].

Figure 7 shows the deviations of the empirical coverage probabilities from the nominal coverage probabilities. We see that the bootstrap confidence intervals from the computational mechanics bootstrap perform extremely well for all three measures, either slightly over-covering or matching the desired coverage probability. The bootstrap confidence intervals from the Markov model bootstrap, however, have 0% coverage at all of the nominal coverages considered. This is again due to a systematic bias in the estimates of $h_{\mu },E,$ and $C_{\mu }$ while using a Markov model to approximate a process that is not Markovian. While the poor performance of the Markov model bootstrap is unsurprising, it does highlight the dangers of assuming a simple Markovian model when investigating systems for which the Markov assumption is not reasonable a priori, even with relatively long time series.

#### 3.1.4. Simple Nonunifilar Source

As our last example, we consider the simple unifilar source [54]. The simple nonunifilar source is neither Markov of any order nor finitary (that is, it has an infinite number of causal states). Figure 2 shows the ϵ-machine architecture for the simple nonunifilar source. The simple nonunifilar source has a unique start state labeled A transitioned to on a 0, and all subsequent 1s transition down a chain of infinitely many causal states. Thus, the simple nonunifilar source is out-of-class for both the Markov model (it is not Markov) and computational mechanics (it is not finitary) bootstraps. Despite this, an ϵ-machine with sufficiently many causal states may be able to approximate the simple nonunifilar source sufficiently well to give an approximate confidence distribution for sufficiently long time series.

Figure 8 shows the deviations of the empirical coverages from the nominal coverages for the simple nonunifilar source. For the shorter time series of length T= 1000, the bootstrap confidence intervals for both the Markov model and computational mechanics bootstraps result in undercoverage for $h_{\mu }$ and E, and have 0% coverage for $C_{\mu }$ for both T= 1000 and T= 10,000. However, for the longer time series of length T= 10,000, the bootstrap confidence intervals from the computational mechanics bootstrap attain the desired coverage for both $h_{\mu }$ and E, while the bootstrap confidence intervals from the Markov model bootstrap continue to undercover. Thus, despite the simple unifilar source being out-of-class relative to the assumptions of the CSSR algorithm used in the computational mechanics bootstrap, the bootstrap confidence intervals from the computational mechanics bootstrap still perform well at covering both $h_{\mu }$ and E given a sufficiently long time series from the simple nonunifilar source.

### 3.2. Twitter Data

The preceding examples demonstrated the inferential properties of both the Markov model and computational mechanics bootstraps on a collection of processes with known properties. We next consider the performance of both bootstrap methods on a diverse set of processes estimated from a collection of 15,000 Twitter users as first described in [9]. The data set consists of the activity patterns of 14,427 Twitter users over a 28-week period. The activity of each user was discretized to active (1)/inactive (0) over 10-minute intervals based on whether the user posted one or more (1) or no (0) tweets in a given interval from 9 AM to 10 PM. At the 10-minute temporal resolution, this results in 78 observations per-day across 196 days, or T= 15,288 total observations per-user. See [9] for additional details regarding the data set.

For each of the users, we treat the ϵ-machine inferred from the original data set as the the ground truth, and generate a new times series of length T= 15,288, which we then use to estimate both a Markov model and ϵ-machine. From the estimated models, we then generate bootstrap confidence distributions and check whether the ground truth values of $h_{\mu }$, E, and $C_{\mu }$ fall in their corresponding confidence intervals.

Figure 9 shows deviation of the empirical coverage probability from the nominal coverage probability for the 14,427 confidence intervals. We see that, for $h_{\mu }$, the percentile bootstrap confidence intervals for the computational mechanics bootstrap perform well, slightly overcovering for smaller nominal coverage probabilities and slightly undercovering for larger coverage probabilities. The other confidence intervals undercover for coverage probabilities typically used. For both E and $C_{\mu }$, all four methods tend to undercover for nominal coverage probabilities greater than 0.6, and drastically so for the confidence intervals that are based on the Markov model bootstrap. The percentile confidence intervals constructed using the computational mechanics bootstrap perform the best, while all confidence intervals based on the Markov model bootstrap perform poorly, undercovering by as much as 40% when a 99% coverage probability is used.

We next turn to investigate whether the undercoverage of the computational mechanics bootstrap is associated with properties of the underlying ϵ-machines. In the ideal case, the coverage probability should not vary with the structure of the underlying ϵ-machine. However, despite the fact that each ϵ-machine was inferred using 15,288 observations, the effective number of observations differs from process-to-process. Two of the main properties that contribute to the effective number of observations are the number of transitions in the underlying ϵ-machine (the larger the number of transitions, the fewer observations available to estimate each transition) and the overall entropy H[X] of the process (if the process almost never or almost always emits 0s, but in a structured way, then the structure of the ϵ-machine will be harder to infer). To this end, we perform a nonparametric logistic regression to estimate the probability that each of the true measures is covered by the 95% confidence interval using the number of transitions *n* in the ϵ-machine and the marginal entropy h=H[X] for each of the users,
(22)$$\begin{array}{c} P(\mathrm{Covered}=1|N=n,H=h)=\mathrm{logit}(g(n,h)). \end{array}$$
where *g* is an arbitrary function estimated using thin plate regression splines via the mgcv package [55].

Figure 10 shows the contour plots for the estimated regression surfaces for each of the measures. Each point in a plot corresponds to one of the 14,427 ϵ-machines, with a green point that corresponds to an ϵ-machine, for which the true measure was captured by the 95% confidence interval and a red point when not. The contour lines show the probability of coverage as a function of the number of transitions and marginal entropy H[X]. We see that, for $h_{\mu }$, the coverage probability does not vary greatly, but does decline for ϵ-machines with relatively low entropies. This agrees with the results from the simulation study, where we saw that the confidence intervals for $h_{\mu }$ were fairly robust to the underlying structure of the ϵ-machine.

However, the coverage for excess entropy and statistical complexity varies greatly depending on the structure of the ϵ-machine and its underlying activity rate, with only those ϵ-machines with a fairly small number of transitions attaining the desired coverage probability, and the coverage probability dropping off very quickly as either the marginal entropy decreases or the number of transitions increases. This effect is starkest for the coverage for statistical complexity, with the coverage dropping from 95% (the desired rate) to as low as 15%, even with a small number of transitions for sufficiently small marginal entropy. Like in the simulation study, this highlights the difficulty of estimating both the excess entropy and statistical complexity.

## 4. Discussion

We used CSSR to estimate the ϵ-machine used in the computational mechanics bootstrap, but other ϵ-machine estimators are available and may perform better. For example, it is known that outputs from finitary stochastic processes, when corrupted by noise, often result in nonfinitary stochastic processes, and an extension to CSSR using robust causal states has been proposed to address this issue [32]. Similarly, there is a folk wisdom amongst users of CSSR that, while consistent in the limit of infinite data, CSSR can result in overly complex ϵ-machines due to certain heuristics used in the reconstruction process, such as choosing to merge a split-history into the state with the largest *p*-value [33]. In [33], the authors show that using the general approach behind CSSR while also guarenteeing that the final ϵ-machine has as few states as possible, given all possible splits and merges, is an NP-hard problem, and propose approximating a solution via integer programming. Further approaches along these lines are possible, for example by treating the global significance level used in the state splitting as a tuning parameter, or apportioning the global significance level among each of the splits and merges in a sequential fashion in order to ensure an overall probability of at most one false split or merge. Any of these methods might endow the confidence distributions derived from the computational mechanics bootstrap with even more desirable inferential properties and should be considered.

We have also only focused on a single method of bootstrapping, namely a type of parametric bootstrap where we treat the estimated ϵ-machine as the approximation to the underlying process, and then bootstrap accordingly. However, nonparametric bootstraps might also be used. For example, the block bootstrap [56] or its variants [57,58] could be used to generate the bootstrap time series from which the bootstrap ϵ-machines are inferred. Similarly, we have only considered two possible types of bootstrap confidence distributions, namely the percentile and bias-corrected bootstrap confidence distributions. Another simple, at least in principle, bootstrap confidence distribution is the confidence distribution that is derived from the bias-corrected and accelerated (BCa) bootstrap confidence interval [19,59,60]. However, the additional acceleration parameter must be estimated from the data, and reasonable estimates for dependent data are non-obvious. Yet another alternative would be a double-bootstrap [61], where the bootstrapped time series are themselves bootstrapped in order to adjust nominal coverage probabilities to more closely achieve the desired levels.

Finally, we have focused on constructing confidence distributions for the information- and computation-theoretic properties of a stochastic process. A more direct approach would be to construct a confidence set for the overall ϵ-machine itself, rather than its properties. Such a confidence set would provide a frequentist analog to the Bayesian posterior distributions developed in [16] for ϵ-machines. This is another instance where the computational mechanics bootstrap might be used, by generating a bootstrap distribution of ϵ-machines, and then using some notion of distance, for example, the variational distance between stochastic processes, to define a data-depth *p*-value [62] that could be used to construct a confidence distribution over the class of all possible ϵ-machines. Given the difficulty that is encountered with constructing confidence distributions for statistical complexity, it is likely that the direct application of the computational mechanics bootstrap, as presented in this paper, would require refinement for this purpose.

## 5. Conclusions

We have developed a method, the computational mechanics bootstrap, for constructing bootstrap confidence distributions for information- and computation-theoretic properties of discrete-time, discrete-state stochastic processes. The resulting confidence distributions can be used for point estimation, interval estimation, and hypothesis testing. We have seen that the computational mechanics bootstrap generally outperforms a Markov model bootstrap, especially when the underlying process is non-Markovian and a sufficiently long time series is available for inference. However, we have also seen that certain information-dynamic properties are easier to make inferences about than others, and the ease of inference depends on the underlying structure of the stochastic process. For the example processes considered, the entropy rate and excess entropy of a process can be reliably inferred with moderately long time series, even for a process (the simple nonunifilar source) that is non-finitary and, thus, outside the class of processes for which the CSSR algorithm is guarenteed to converge. The statistical complexity of a process, however, tends to be more difficult to infer, especially for a process with infinitely many causal states.

With the investigation of the models derived from the activity of users on Twitter, we have seen that the computational mechanics bootstrap performs fairly well at making inferences about entropy rates, but underperforms for both excess entropy and statistical complexity. However, in all cases, it performs better than the Markov model bootstrap, and Markov models are amongst the most common models used to make inferences about information-theoretic quantities. Therefore, we urge caution when using Markov models to estimate such properties, especially since estimating an ϵ-machine is generally no more difficult than estimating a Markov model, the desired properties are readily calculable from the estimated ϵ-machine, and the resulting estimators have better inferential properties for a wide class of processes.

## Figures and Tables

**Figure 1 entropy-22-00782-f001:**
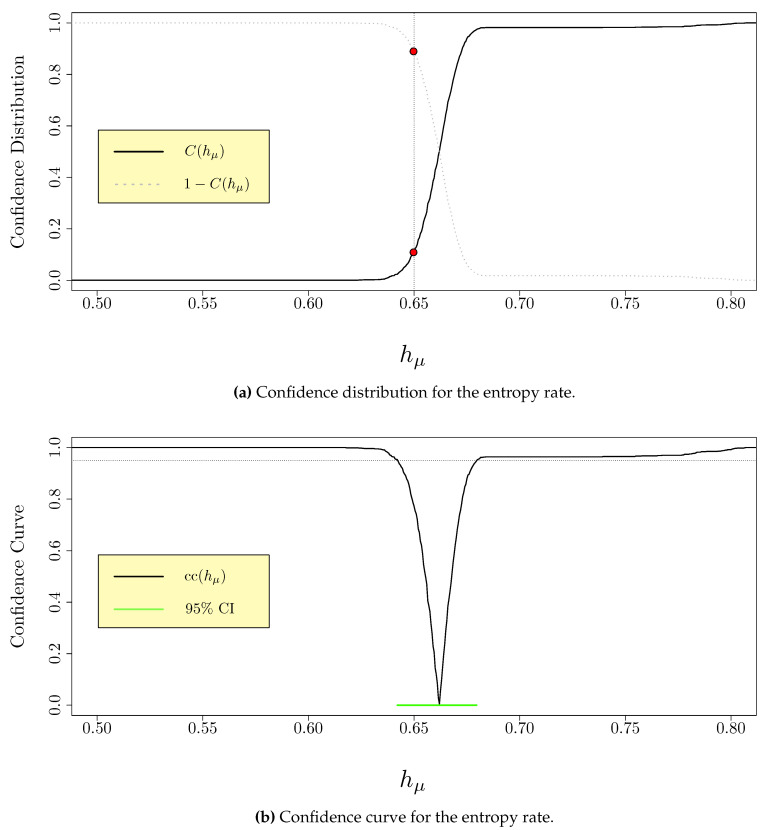
Example confidence distribution and confidence curve for the entropy rate $h_{\mu }$ of a process. (**a**) The confidence distribution $C(h_{\mu })$ (solid black) and complementary confidence distribution $1-C(h_{\mu })$ (dashed grey) are shown, as well as the *p*-values for the right-sided and left-sided tests with null value $h_{\mu }=0.65$, given by C(0.65)≈0.114 and 1−C(0.65)≈0.886, respectively. The *p*-value for the two-sided test at $h_{\mu }=0.65$ is 2min{0.114,0.886}=0.228. (**b**) The confidence curve for $h_{\mu }$, with the 95% confidence interval (0.642,0.680) shown in green.

**Figure 2 entropy-22-00782-f002:**
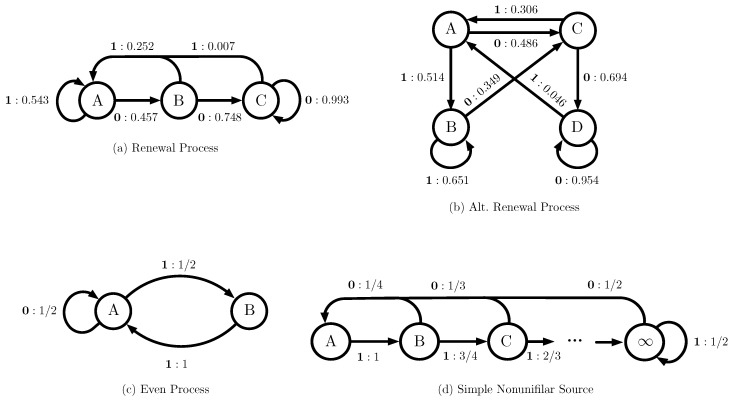
The ϵ-machine representations of the stochastic processes considered in the simulation study in Section 3.1. The arrows are decorated with x:p, where x is the emission symbol and *p* is the probability of that emission symbol given the current causal state. (**a**) A renewal process with three states. (**b**) An alternating renewal process with four states, equivalent to a second-order Markov model. (**c**) The even process, a sophic process that is not Markov for any order. (**d**) The simple unifilar source, a non-finitary stochastic process.

**Figure 3 entropy-22-00782-f003:**
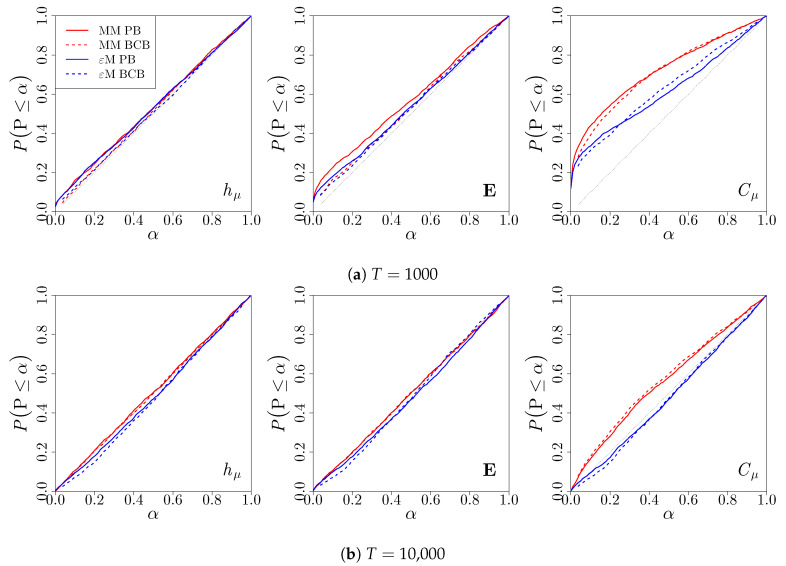
Empirical distribution of two-sided *p*-values for the hypotheses (17) for the three-state renewal process. Bootstrap *p*-values were constructed using a Markov model (red) or *ϵ*-machine (blue) simulator with percentile bootstrap (solid) or bias-corrected bootstrap (dashed) confidence distributions. The proportions of *p*-values less than or equal to a are reported. (**a**) *S* = 2000 time series of length *T* = 1000 were used. (**b**) *S* = 2000 time series of length *T* = 10,000 were used.

**Figure 4 entropy-22-00782-f004:**
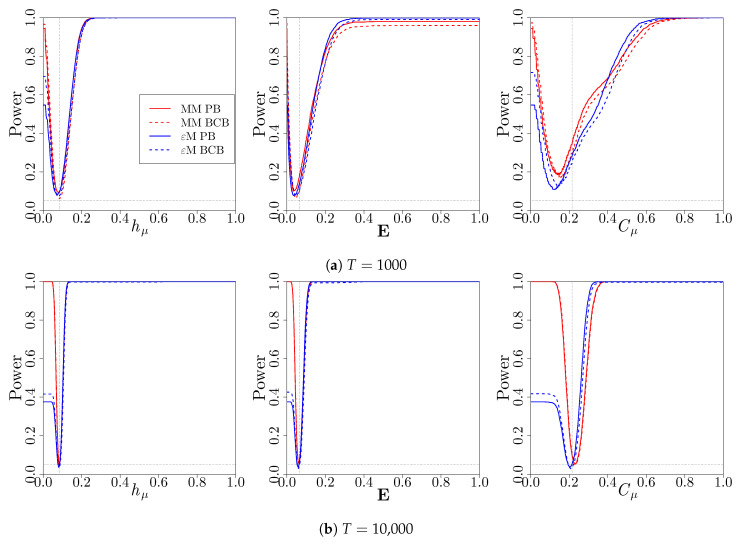
Power of the two-sided hypothesis tests for the hypotheses (17) for the three-state renewal process with *α* = 0.05 as a function of the null value *θ*_0_. Bootstrap *p*-values were constructed using a Markov model (red) or *ϵ*-machine (blue) simulator with percentile bootstrap (solid) or bias-corrected bootstrap (dashed) confidence distributions. The proportions of *p*-values less than or equal to a are reported. The horizontal dashed lines indicate a power of 0.05, the significance level used for the tests. The vertical dashed lines indicate the true values of the measures listed in Table 1.

**Figure 5 entropy-22-00782-f005:**
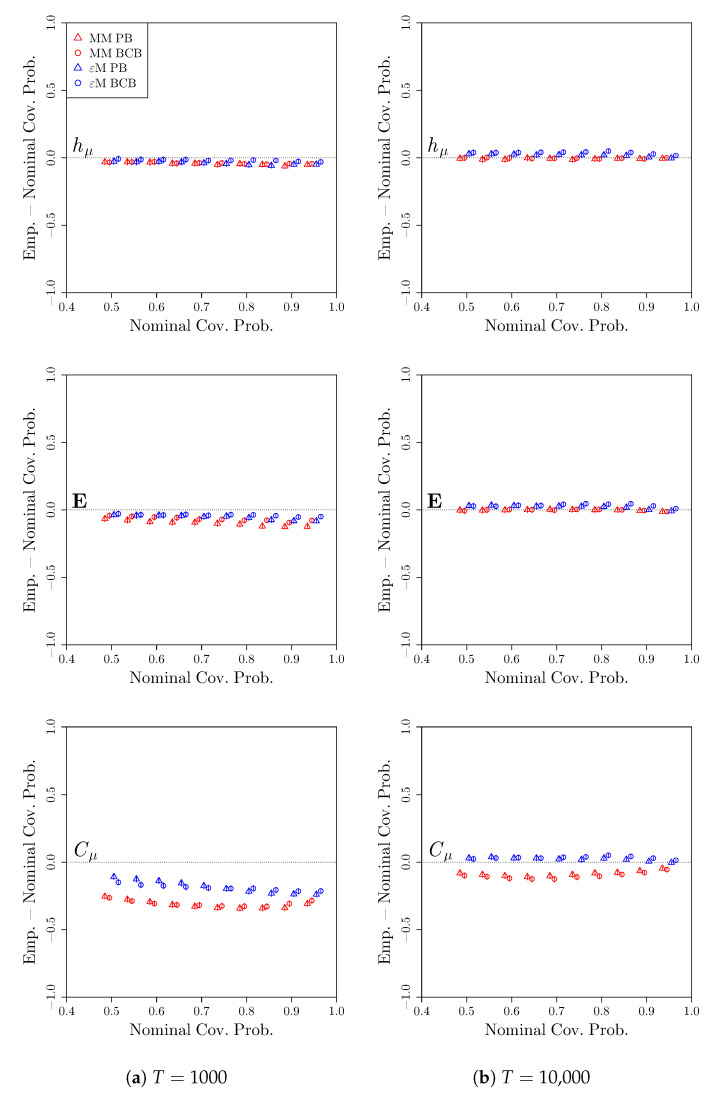
Deviation of empirical coverage probabilities from nominal coverage probabilities for the three-state renewal process. Bootstrap confidence intervals were constructed using a Markov model (red) or *ϵ*-machine (blue) simulator with percentile bootstrap (triangle) or bias-corrected bootstrap (circle) confidence distributions. The estimated coverage deviations and pointwise 95% confidence intervals for the coverage deviations are reported. *S* = 2000 time series of length (**a**) *T* = 1000 and (**b**) *T* = 10,000 were used.

**Figure 6 entropy-22-00782-f006:**
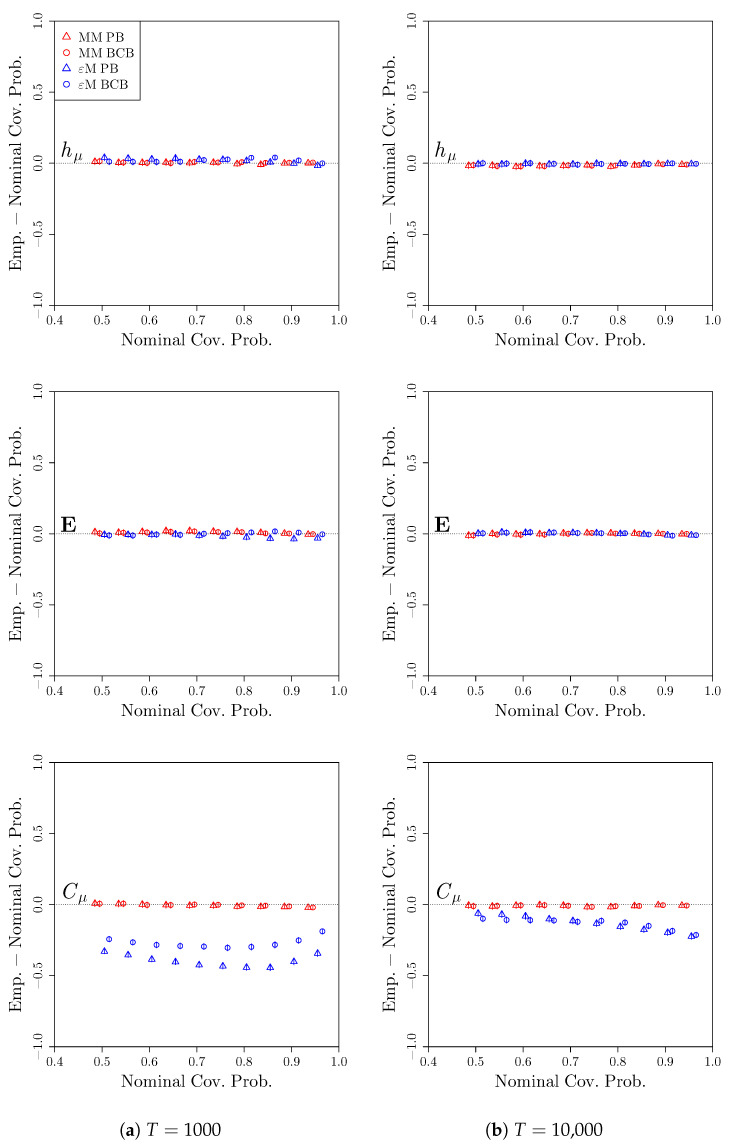
Deviation of empirical coverage probabilities from nominal coverage probabilities for the 4-state alternating renewal process. Bootstrap confidence intervals were constructed using a Markov model (red) or *ϵ*-machine (blue) simulator with percentile bootstrap (triangle) or bias-corrected bootstrap (circle) confidence distributions. The estimated coverage deviations and pointwise 95% confidence intervals for the coverage deviations are reported. *S* = 2000 time series of length (**a**) *T* = 1000 and (**b**) *T* = 10,000 were used.

**Figure 7 entropy-22-00782-f007:**
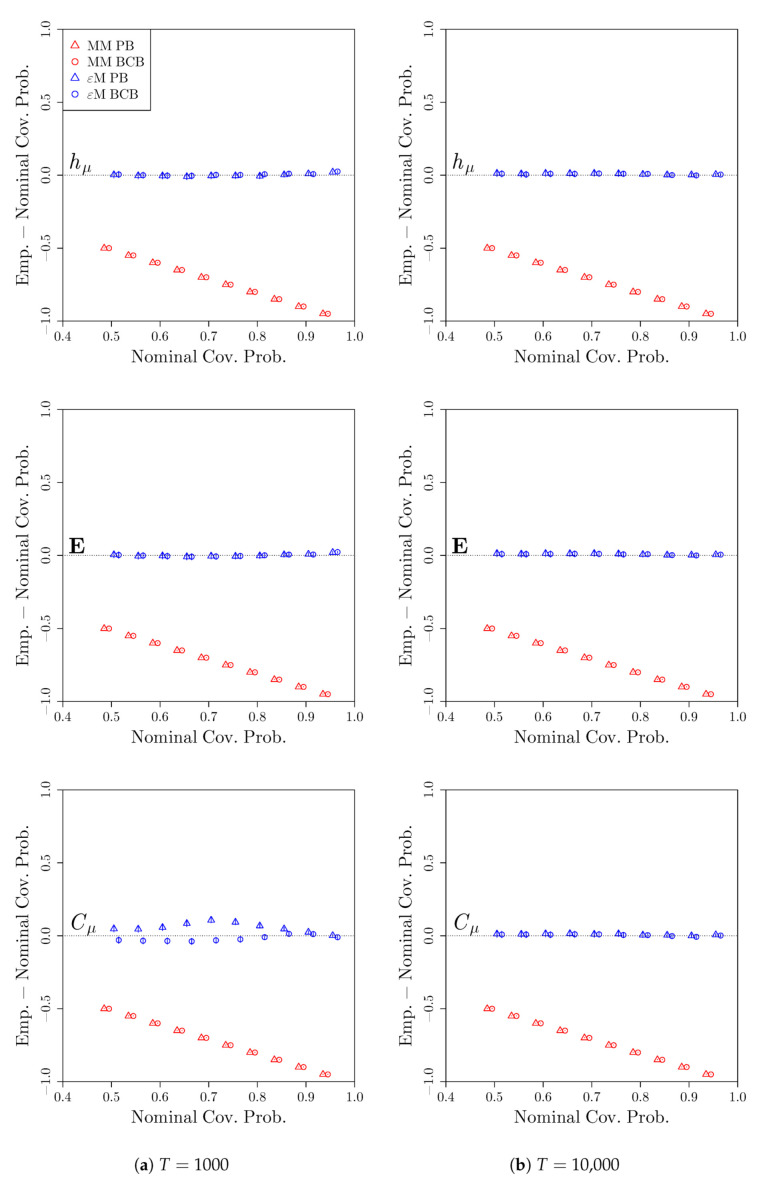
Deviation of empirical coverage probabilities from nominal coverage probabilities for the even process. Bootstrap confidence intervals were constructed using a Markov model (red) or *ϵ*-machine (blue) simulator with percentile bootstrap (triangle) or bias-corrected bootstrap (circle) confidence distributions. The estimated coverage deviations and pointwise 95% confidence intervals for the coverage deviations are reported. *S* = 2000 time series of length (**a**) *T* = 1000 and (**b**) *T* = 10,000 were used.

**Figure 8 entropy-22-00782-f008:**
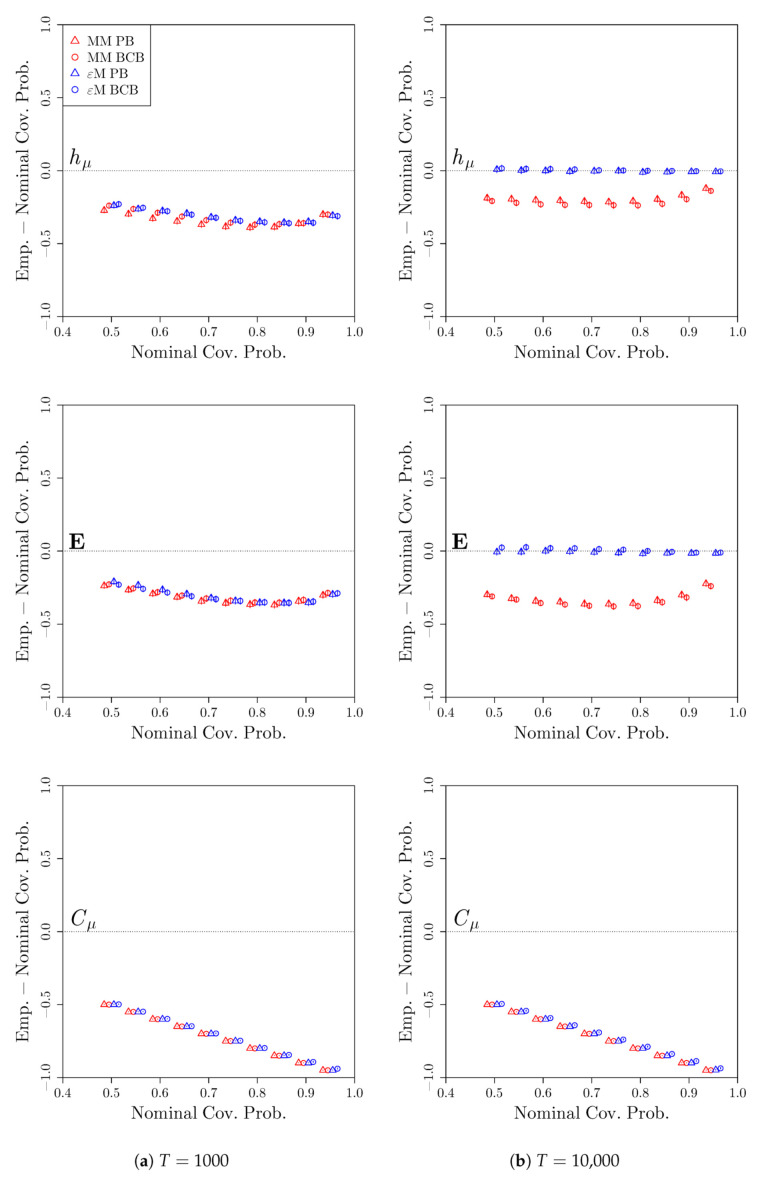
Deviation of empirical coverage probabilities from nominal coverage probabilities for the simple nonunifilar source. Bootstrap confidence intervals were constructed using a Markov model (red) or *ϵ*-machine (blue) simulator with percentile bootstrap (triangle) or bias-corrected bootstrap (circle) confidence distributions. The estimated coverage deviations and pointwise 95% confidence intervals for the coverage deviations are reported. *S* = 2000 time series of length (**a**) *T* = 1000 and (**b**) *T* = 10,000 were used.

**Figure 9 entropy-22-00782-f009:**
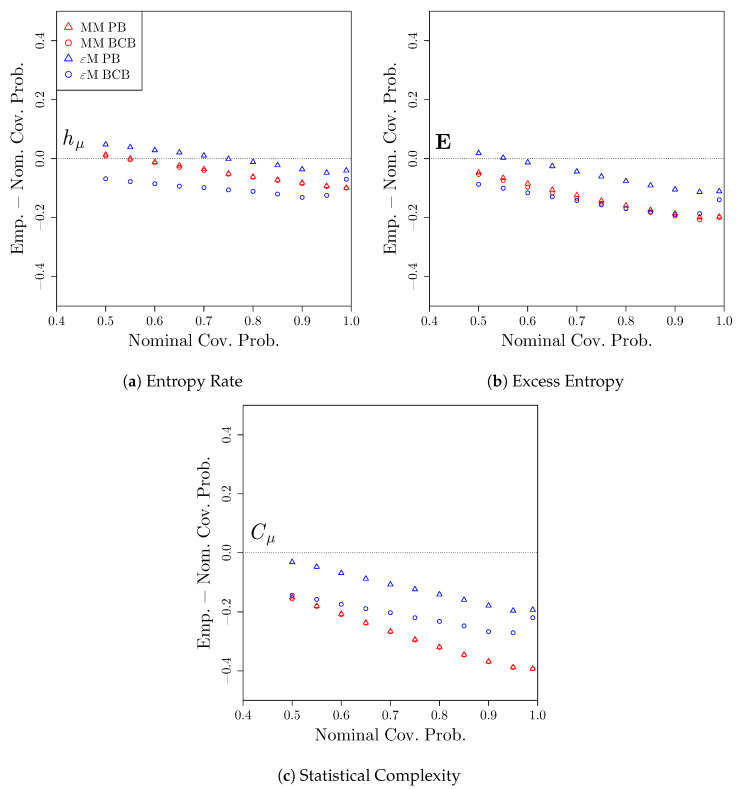
Deviation of empirical coverage probabilities from nominal coverage probabilities for the processes derived from the Twitter data set. Bootstrap confidence intervals were constructed using a Markov model (red) or e-machine (blue) simulator with percentile bootstrap (triangle) or bias-corrected bootstrap (circle) confidence distributions.

**Figure 10 entropy-22-00782-f010:**
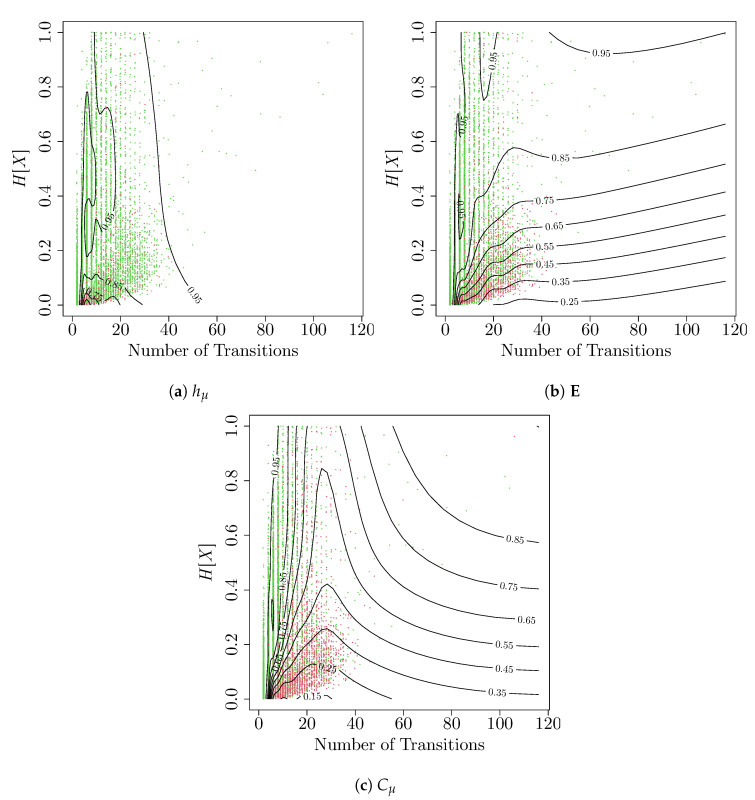
The estimated coverage probabilities for the 95% percentile confidence intervals constructed from the computational mechanics bootstrap as a function of the number of transitions and marginal entropy *H*[*X*] for (**a**) *h_μ_*, (**b**) **E**, and (**c**) *C_μ_*. Each green (red) point corresponds to a single user whose measure was captured (missed) by the 95% confidence interval.

**Table 1 entropy-22-00782-t001:** The entropy rate $h_{\mu }$, excess entropy E, and statistical complexity $C_{\mu }$ for the four processes considered in the simulation study. Each measure is computed from the true ϵ-machine and reported in bits to eight decimal places.

Process	$h_{\mu }$ (Bits)	*E* (Bits)	$C_{\mu }$ (Bits)
3-State Renewal	0.08560820	0.06376713	0.21520538
4-State Alternating Renewal	0.39456572	0.19801359	0.98714503
Even	2/3	0.91829583	0.91829583
Simple Nonunifilar Source	0.67786718	0.14723194	2.71146872

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
