# Peer review of "Discrete Information Dynamics with Confidence via the Computational Mechanics Bootstrap: Confidence Sets and Significance Tests for Information-Dynamic Measures"

_entropy, 2020, doi:10.3390/e22070782_

Round 1

Reviewer 1 Report

This article summarizes a new method for developing confidence sets and significance tests for three information dynamics quantities.

This article proposes what seems like a sensible idea to me.  My only qualm is that when bootstrapping, new samples are generated from the inferred eM.  If you accidentally got the topology wrong, your confidence intervals are harder to believe, which I think the author kind of touches on later on in the article.  Other than that:

-- There are a few terms that I did not know ("pivot", "exact pivot", and "confidence distribution" actually) that the author might want to take more care in explaining (e.g., why is condition (b) regarding confidence distributions sensical?).

-- When using BIC to choose a Markov order, it might make sense to refer to a Strelioff and Crutchfield PRE on choosing Markov orders, if only to cite it for those interested in a more exact method

-- The author says that finding the bidirectional eM and that the spectral method are not easily automatable, but I beg to disagree!  I actually have code, which I would be happy to provide upon request as long as it is not widely shared :)

-- Some of the figures didn't appear in my version of the manuscript; just something to check before publishing.

Overall, I was happy to see more research in this direction!

Author Response

Response to Reviewer 1:

Thank you very much for your insights and suggestions. I believe they have greatly improved my paper. I have addressed each point from your review below:

"My only qualm is that when bootstrapping, new samples are generated from the inferred eM. If you accidentally got the topology wrong, your confidence intervals are harder to believe, which I think the author kind of touches on later on in the article."

I agree it is counterintuitive that the bootstrap could still work even in those cases where the wrong \epsilon-machine (eM) topology is inferred from a given time series. However, this is not so different from the general situation of bootstrapping, where the sample, rather than the population, is used to approximate the sampling distribution for a statistic. As long as the topology of the inferred eM is "not too far off" compared to the true topology, the P-values, confidence intervals, etc., can still be approximately valid. More work needs to be done to see if this idea extends to constructing confidence sets for the eM topology itself.

"-- There are a few terms that I did not know ("pivot", "exact pivot", and "confidence distribution" actually) that the author might want to take more care in explaining (e.g., why is condition (b) regarding confidence distributions sensical?)."

Thank you for the suggestion to add more clarification about these concepts. I have expanded the first paragraph of Section 2.5 to further flesh out these ideas.

"-- When using BIC to choose a Markov order, it might make sense to refer to a Strelioff and Crutchfield PRE on choosing Markov orders, if only to cite it for those interested in a more exact method"

I have added the reference to Strelioff and Crutchfield's 2007 paper in PRE.

"-- The author says that finding the bidirectional eM and that the spectral method are not easily automatable, but I beg to disagree! I actually have code, which I would be happy to provide upon request as long as it is not widely shared :)"

I would be very interested in using that code if you can make it available to me! I have been unsuccessful in applying the methods from "Exact Complexity: The Spectral Decomposition of Intrinsic Computation" to work automatically with large eMs, especially when (a) the required matrix is not diagonalizable and (b) there are a large number of mixed states, such that computing the matrix decomposition results in numerical errors that swamp the exact results.

I have removed that sentence from the manuscript.

"-- Some of the figures didn't appear in my version of the manuscript; just something to check before publishing."

I apologize: the figures appear correctly in all of the PDF viewers I tried on macOS (Preview, Skim, Chrome, Safari, etc.), but apparently cause an error in the Adobe suite of PDF viewers. I have fixed Figures 1 and 2 so that they should now appear correctly.

Sincerely,
David Darmon

Reviewer 2 Report

Major/minor comments:

  1. Re-write the abstract to reflex the content and try to reduce the new abstract.
  2. What is the motivation of this paper?. The author should write some merits and aims around his paper. 
  3. Some equations along the paper has duplicate number. Please, see Equations 1,2,4,5,10,11,12,13,17,18,19,20,21,22-25.
  4. Write an algorithm for the computational mechanics bootstrap.
  5. Where are figures 1 and 2 along the paper?
  6. Write the discussion Section in a separate Section.
  7. Write the conclusion Section in a separate Section including the important scientific results only. 

Author Response

Response to Reviewer 2:

Thank you very much for your insights and suggestions. I believe they have greatly improved my paper. I have addressed each point from your review below:

1. Re-write the abstract to reflex the content and try to reduce the new abstract.

I have condensed the abstract and made it more concisely address the contents of the paper.

2. What is the motivation of this paper?. The author should write some merits and aims around his paper.

The motivation of this paper is to construct hypothesis tests and confidence sets for general information- and computation-theoretic measures. I have added a sentence to the paragraph beginning on line 55 to make this point more clear.

3. Some equations along the paper has duplicate number. Please, see Equations 1,2,4,5,10,11,12,13,17,18,19,20,21,22-25.

Thank you for pointing out my oversight. I have corrected these equations so they only have a single number per-equation.

4. Write an algorithm for the computational mechanics bootstrap.

I have added a box summarizing the computational mechanics bootstrap on line 195.

5. Where are figures 1 and 2 along the paper?

I apologize: the figures appear correctly in all of the PDF viewers I tried on macOS (Preview, Skim, Chrome, Safari, etc.), but apparently cause an error in the Adobe suite of PDF viewers. I have fixed Figures 1 and 2 so that they should now appear correctly.

6. Write the discussion Section in a separate Section.

7. Write the conclusion Section in a separate Section including the important scientific results only.

6--7: I have separated my discussions and conclusions into separate sections.

Sincerely,
David Darmon